# *Lactobacillus sakei* MJM60958 as a Potential Probiotic Alleviated Non-Alcoholic Fatty Liver Disease in Mice Fed a High-Fat Diet by Modulating Lipid Metabolism, Inflammation, and Gut Microbiota

**DOI:** 10.3390/ijms232113436

**Published:** 2022-11-03

**Authors:** Huong Thi Nguyen, Mingkun Gu, Pia Werlinger, Joo-Hyung Cho, Jinhua Cheng, Joo-Won Suh

**Affiliations:** 1Interdisciplinary Program of Biomodulation, Myongji University, Yongin 17058, Korea; 2Myongji Bioefficacy Research Center, Myongji University, Yongin 17058, Korea

**Keywords:** *Lactobacillus sakei* MJM60958, NAFLD, probiotics, microbiota, SCFAs

## Abstract

Non-alcoholic fatty liver disease (NAFLD) is a common liver disease with a rapidly increasing number of cases worldwide. This study aimed to evaluate the effects of *Lactobacillus sakei* MJM60958 (MJM60958) on NAFLD in vitro and in vivo. In in vitro tests, MJM60958 significantly inhibited lipid accumulation by 46.79% in HepG2 cells stimulated with oleic acid and cholesterol (OA-C). Moreover, MJM60958 showed safe and probiotic characteristics in vitro. In the animal study, MJM60958 administration in a high-fat diet-induced NAFLD mouse model significantly reduced body weight and liver weight, and controlled aspartate aminotransferase (ALT), aspartate transaminase (AST), triglyceride (TG), urea nitrogen (BUN), and uric acid (UA) levels in the blood, which are features of NAFLD. Further, treatment with MJM60958 also reduced steatosis scores in liver tissues, serum leptin and interleukin, and increased serum adiponectin content. Moreover, administration of MJM60958 resulted in a significantly decreased expression of some genes and proteins which are related to lipid accumulation, such as fatty acid synthase (FAS), acetyl-CoA carboxylase (ACC), and sterol regulatory element-binding protein 1 (SREBP-1), and also upregulated genes and protein expression of lipid oxidation such as peroxisome proliferator-activated receptor alpha (PPARα) and carnitine palmitoyltransferase 1a (CPT1A). Administration of MJM60958 increased the relative abundance of specific microbial taxa such as Verrucomicrobia, which are abundant in non-NAFLD mice, and reduced Firmicutes, which are a major group in NAFLD mice. MJM60958 affected the modulation of gut microbiota and altered the strain profile of short-chain fatty acids (SCFAs) production in the cecum by reduced lactic acid and enhanced acetic acid production. Overall, MJM60958 showed potential as a probiotic that can prevent and treat NAFLD.

## 1. Introduction

Non-alcoholic fatty liver disease (NAFLD) is the main cause of chronic liver diseases and has dramatically increased worldwide over the past decade (the number of NAFLD patients is estimated to be approximately 25% of the global population) [1,2,3]. NAFLD is characterized by the accumulation of fat in hepatocytes exceeding 5% of liver weight without other causes of liver diseases such as viral hepatitis, autoimmune liver disease, and alcohol consumption. It is estimated that approximately 23% of NAFLD patients will progress to non-alcoholic steatohepatitis (NASH), about 10–25% of NASH patients will progress to cirrhosis, and about 2.4–12.8% of cirrhotic patients will progress to liver cancer [4,5]. NAFLD is predicted to become the main indication for liver transplantation by 2030. By 2030, the liver transplantation indicator is predicted highest for NAFLD [6].

Obesity, diabetes, and metabolic syndrome are closely associated with NAFLD [7,8]. However, NAFLD also occurs in non-obese and non-diabetes patients with metabolism disorders [9]. NAFLD begins with an excessive accumulation of fat in the liver mainly derived from dietary fatty acids, de novo lipogenesis, and adipose tissue. The imbalance between the synthesis and oxidation of fatty acid contributes to NAFLD. With fatty acid uptake, hepatic insulin resistance leads to impairment of glucose homeostasis, decreases fatty acid oxidation, and contributes to triglycerides (TG) accumulation. In addition, the de novo lipogenesis pathway comprises glycolysis biosynthesis of saturated fatty acid followed by desaturation and TG formation. Key rate-limiting enzymes in this process include glucokinase and liver-type pyruvate kinase in glycolysis, acetyl-CoA carboxylase (ACC), and fatty acid synthase (FAS). ACC catalyzes a critical rate-limiting step in fatty acid biosynthesis and controls fatty acid oxidation by the synthesis of malonyl-CoA, an inhibitor for carnitine palmitoyl transferase 1 (CPT1). Sterol regulatory element binding protein 1c (SREBP-1c) is a member of SREBP that is mainly a transcription factor for lipogenesis in NAFLD. Further, peroxisome proliferator-activated receptor alpha (PPARα) and CPT1A control fatty acid beta-oxidation, a process that shortens fatty acids into acetyl-CoA. Hallmarks of the progression of NAFLD to NASH are inflammation and liver injury, with the release of cytokines and chemokines, such as IL-1β and tumor necrosis factor α (TNF-α) [10,11].

Current NAFLD management includes diet, and exercise to control weight and reduce underlying metabolic syndrome [12]. The development of effective drugs to prevent and treat NAFLD is an area of interest to scientists. Moreover, gut microbiome dysbiosis leads to NAFLD development due to dysfunction of the gut–liver axis, and probiotics also can protect the liver via the gut–liver axis [13].

Probiotics are living microorganisms that contribute health benefits to the host when consumed in adequate amounts [14]. Lactobacillus and Bifidobacterium species are safe and by far the most commonly used probiotics [15]. Among Lactobacillus species, an important probiotic is *Lactobacillus sakei*, which has been recognized for many functions such as anti-obesity [16], anti-colitis [17], anti-oxidant [18], immunostimulatory [19], alleviation of atopic [20], antimicrobial [21], anti-diabetic and anti-melanogenic [22] activities. However, until now there is no research evaluating the effect of *Lactobacillus sakei* on NAFLD. In this study, *Lactobacillus sakei* MJM60958 was screened for its high anti-lipid accumulation in HepG2 cells, and the effect of MJM60958 on the NAFLD was evaluated in a mice model fed with a high-fat diet (HFD). In addition, to be developed as a probiotic, MJM60958 was tested with several safety tests including hemolytic activity, D-lactate production, bile salt deconjugation, biogenic amine production, antimicrobial activity, mucin degradation, antibiotic susceptibility, oro-gastrointestinal transit assay, and cell adhesion assay.

## 2. Results

### 2.1. Screening of LAB Strains Capable of Reducing Lipid Accumulation and Cytotoxicity of Selected LAB Strain on HepG2 Cells

The activity reduced lipid accumulation in NAFLD in HepG2 cells of 15 probiotic strains isolated from fermented food, as shown in Table 1. Among 11 strains that showed effectiveness, 3 strains with higher activity than simvastatin were *Lactobacillus gasseri* MJM61024, *Lactobacillus brevis* MJM60386, and *Lactobacillus sakei* MJM60958. The most effective strain, MJM60958, reduced lipid synthesis by 46.79% compared with the OA-C group and was chosen for further study (Table 1, Figure 1A,B). Moreover, the effect of MJM60958 on the cell viability of HepG2 cells also was conducted using an MTT assay. MJM60958 treated at both concentrations of 10^8^ and 10^9^ CFU/mL showed no toxicity and slightly improved HepG2 cell viability compared with the OA-C group (Figure 1C).

### 2.2. Safety Assessment of MJM60958

According to the cut-off values recommended by EFSA (2012), MJM60958 was tested with nine types of antibiotics, and the MIC values are listed in Table 2. Strain MJM60958 was susceptible to all antibiotics. Moreover, hemolytic activity, mucin degradation activity, D-lactate production, bile salt deconjugation, or bioamine production was not exhibited by MJM60958 (Table 2).

The adherent ability of MJM60958 and LGG to HT-29 cells was determined by the ratio between the number of cells that remained attached to the HT-29 monolayer after washing three times and the initial cell number. The adherence rate of MJM60958 (5.09%) was significantly higher than that of LGG (3.19%) (Table 2).

### 2.3. OGI Transit Assay

The tolerance of the MJM60958 strain to the digestion process was examined by exposure to OGI transit stresses, and the survival of MJM60958 is shown in Table 3. Without stress, the cell viability of LGG and MJM60958 was not changed. The log unit CFU at the end of the OGI assay was significantly reduced by 1.33 log unit CFU (*p* < 0.05) and 1.89 log unit CFU (*p* < 0.001) compared with the initial counts for LGG and MJM60958, respectively. However, the log unit CFU of MJM60958 after the OGI process was still higher than 7 (Table 3).

### 2.4. Antibacterial Activity of MJM60958

The results in Table 4 show that MJM60958 strains had antibacterial activity against all tested pathogens. MJM60958 strongly inhibited *Salmonella gallinarum* KCTC 2931, *Escherichia coli* O1 KCTC 2441, *Salmonella choleraesuis* KCTC 2932, and *Pseudomonas aeruginosa* KCCM 11802. It also moderates against *Escherichia coli* K99, *Escherichia coli* O138, *Escherichia coli* ATCC25922, and *Salmonella typhi* KCTC 2514. Similarly, LGG also had strong or moderate antibacterial activity against these pathogens (Table 4).

### 2.5. Animal Study

#### 2.5.1. Effect of MJM60958 on the Food Intake, Body Weight, and Organ Weights

Normal diets and high-fat diets were fed to each group for 12 weeks, and food intake and body weight were measured during the 12 weeks. In the first two weeks, no difference in food intake between groups was observed, but the food intake of the control group with a normal diet was higher than groups on HFD from week 3. However, there was no significance among HFD, Silymarin, and MJM60958 groups which were on HFDs (Figure 2A). In addition, body weight significantly increased in all groups as time passed (Figure 2B). The normal diet control group had the highest level of food consumption but with low calories, and the body weight of mice in this control group was still the lowest (30.35 ± 1.86 g at week 12). On the other hand, in the HFD group with high calories, the body weight of mice was highest (42.93 ± 2.36 g). However, body weight in silymarin and MJM60958 (10^9^) groups strongly decreased to 33.78 ± 2.45 g and 34.14 ± 1.83 g, respectively, compared with the HFD after 12 weeks (*p* < 0.0001). Further, with lower probiotic treatment, MJM60958 (10^8^) just slightly decreased the body weight of mice (Table 5). Furthermore, there was no significant difference in kidney weight and epidydimal fat weight between groups fed with HFD. By contrast, liver weight and liver weight/body weight in HFD were significantly reduced by MJM60958 treatment.

#### 2.5.2. Effect of MJM60958 on Histopathology of the Liver Sections

To assess the ameliorating effect of MJM60958 on hepatic histopathological changes in mice, the H&E staining of liver tissues was analyzed. (Figure 3A–D). The control group did not show the presence of ballooned liver cells. However, the number of ballooned hepatocytes with lipid droplets dramatically increased in the HFD group, revealing that HFD induced damage in liver tissues with severe steatohepatitis. After oral administration with LAB, the quantity of ballooned hepatocyte significantly reduced, especially at the high-dose LAB treatment steatosis grade of MJM60958 (10^9^). The results of steatosis grading in Figure 3F demonstrate that the percentage of fat within the hepatocytes is severe in the HFD group (grade 3). In the LAB treatment, the fat content in the liver significantly decreased to between mild and moderate levels, with similar levels in the silymarin group (grade 1 and grade 2).

#### 2.5.3. Effect of MJM60958 on Serum Levels of ALT, AST, TG, TCHO, CRE, UA, Leptin, Adiponectin, IL-1β, and TNF-α

HFD significantly increased ALT, AST, TG, and TCHO levels in serum by 2.39-, 1.33-, 1.47-, and 1.56-fold, respectively, compared with the control group. Moreover, HFD notably decreased BUN (*p* < 0.01), and slightly regulated serum UA and CRE compared with the control group (Figure 4). LAB treatments with both low (10^8^ CFU) and high (10^9^ CFU) doses showed significantly reduced levels of ALT (*p* < 0.001 and *p* < 0.0001), AST (*p* < 0.0001), UA (*p* < 0.05), and marked increases in BUN (*p* < 0.05). In addition, MJM60958 (10^8^) and MJM60958 (10^9^) also slightly decreased serum TCHO and slightly induced the serum CRE index. However, only the high-dose treatment of LAB significantly reduced serum TG (*p* < 0.0001) compared with HFD (Figure 4).

Compared with the control group, the HFD group had significantly elevated leptin levels in serum (*p* < 0.001) and suppressed serum adiponectin (*p* < 0.01). In contrast, the MJM60958 (10^9^) showed a significant reduction (*p* < 0.0001) in serum leptin and enhanced serum adiponectin (*p* < 0.0001), whereas low-dose LAB treatment MJM60958 (10^8^) only significantly raised adiponectin level (*p* < 0.0001) compared with HFD (Figure 4H,I). Previous studies revealed that tumor necrosis factor-alpha (TNF-α) and interleukin-1-beta (IL-1β) are key mediators in the pathogenesis of NAFLD [23]. In our study, HFD significantly increased IL-1β (*p* < 0.05) and slightly increased TNF-α in serum, but serum TNF-α and serum IL-1β were considerably reduced in the MJM60958 groups (*p* < 0.05) by the LAB supplement, especially in the high-dose group (Figure 4J,K).

#### 2.5.4. Effect of MJM60958 on mRNA Expression of Lipogenesis GENES and Inflammation-Related Genes in Mice Livers

Since a high dose of MJM60958 showed more significant activity than a low dose, the mechanism study was performed only in the high-dose treatment (MJM60958). We evaluated the expression of genes relating to lipogenesis and inflammation function in liver tissue. HFD consumption dramatically upregulated FAS and ACC gene expression by 2.39- and 2.36-fold, respectively, compared with the control group (Figure 5A,B). However, the supplement of MJM60958 significantly downregulated FAS and ACC gene expressions (*p* < 0.05). Moreover, HFD significantly reduced the expression of PPARα and CPT1A energy metabolism-related genes (*p* < 0.0001), whereas MJM60958 dramatically increased gene expression of PPARα and CPT1A (*p* < 0.0001) compared with HFD (Figure 5C,D). Further, both silymarin and MJM60958 showed insignificant regulation of IL-6 compared with HFD (Figure 5E).

#### 2.5.5. Effect of MJM60958 on the Expression of FAS, SREBP-1, and PPARα Proteins in Mice Livers

To assess the effects of MJM60958 on lipid metabolism, the expression of key proteins involved in lipogenesis and fatty acid oxidation was determined (Figure 6).

The protein level of FAS was highly upregulated by HFD (*p* < 0.0001), but significantly downregulated by the treatment of silymarin and MJM60958 (*p* < 0.001 and *p* < 0.0001, respectively) (Figure 6B). Moreover, when treated with HFD, protein expression of SREBP-1 also significantly increased (*p* < 0.05), whereas the level of SCREB1 was found to be significantly reduced by MJM60958 (*p* < 0.05) (Figure 6C). Further, HFD suppressed the protein expression of PPARα compared with the control (*p* < 0.01). Silymarin and MJM60958 treatment strongly enhanced the protein level of PPARα at significant levels of *p* < 0.001 and *p* < 0.01, respectively (Figure 6D).

#### 2.5.6. Composition of Cecal Microbiota and SCFAs Content in Cecal

The effect of MJM60958 on gut microbiota was determined based on the relative abundance of bacteria in the cecum. The composition of the gut microbiota at both phylum and genus levels was analyzed to determine which types of bacteria were affected by MJM60958 in the HFD group. At the phylum level, Firmicutes, Bacteroides, Proteobacteria, and Verrucomicrobia were the main components of the microbiota, accounting for more than 98% of the total. Compared with the control group, an increase in Firmicutes (8.4%) and Actinobacteria (0.3%) and the disappearance of Venrrucomicrobia and Tenericutes were observed in the HFD group. However, MJM60958 treatment significantly reduced Firmicutes (4.6%) and Actinobacteria (0.6%) and improved gut microbiota diversity with the presence of Venrrucomicrobia (3.85%) and Tenericutes (1.033%) compared with HFD (Figure 7A). The abundance of the gut microbiota at the family level was also changed. Compared with the control group, in the HFD group, the abundance of Atopobiaceae, Deferribacteraceae, Christensenallaceace, Clostridiaceae 1, Clostridiales vadinBB60 group, Lachnospiraceae, Peptostreptococcaceae, and Anaeroplasmataceae was increased, and there was decrease in the abundance of Family XIII, Ruminococcaceae, Anaeroplasmataceae and an absence of Akkermansiaceae. With MJM60958 treatment, Akkermansiaceae and Ruminococcaceae were improved by 3.35% and 1.68%, respectively, whereas Clostridiales vadinBB60 group, Clostridiaceae 1, Atopobiaceae, Lachnospiraceae, and Peptostreptococcaceae were reduced by 2.5%, 0.32%, 0.1%, 1.34%, and 0.45%, respectively, compared with HFD (Figure 7B).

GC-MS was used to analyze short-chain fatty acids (SCFAs) content in the feces. However, only lactic acid and acetic acid were detected as present in each group, as shown in Figure 7C,D. The lactic acid content was 1152.1 mg/g in the control group, increased to 1343.21 mg/g in the HFD group, and significantly decreased to 593.5 and 737.95 mg/g in the silymarin and MJM60958 groups, respectively (Figure 7C). Compared with the control group, acetic acid concentration was dramatically reduced by 211.75 mg/g in HFD, whereas silymarin and MJM606958 treatment significantly enhanced the acetic acid by 162.99 and 140.58 mg/g, respectively, compared with the HFD group (Figure 7D).

## 3. Discussion

Non-alcoholic fatty liver disease (NAFLD) is the most common liver disease which affects both adults and children [24]. NAFLD has a complex physiopathology related to insulin resistance, obesity, type 2 diabetes, and cardiovascular diseases [7,8,25,26]. To date, there is no specific pharmacologic treatment for NAFLD, but some medicines can treat the physiopathology of NAFLD, including insulin-sensitizers [27], lipid-lowering agents [28], antioxidant agents [29], and antitumor necrosis factor α agents [30]. However, there are no medications that have been approved for the treatment of NAFLD, despite many clinical trials being conducted. Recently, using probiotics has been considered a potential approach for preventing and improving NAFLD [31,32,33].

MJM60958 was isolated from dongchimi, a Korean traditional fermented radish kimchi, and obtained by our laboratory. It was selected from 15 LAB strains for its strong activity inhibiting lipid accumulation in HepG2 cells and absence of toxicity for hepatocytes (Table 1, Figure 1). In addition, MJM60958 is a potential probiotic with the safety and characteristics of probiotics. D-lactate is produced by some LAB that lead to short bowels and acidosis disease [34,35]. Bile salt deconjugate increased the survival of LAB in the intestinal environment; however, excess bile salt deconjugate is toxic for the human intestinal tract [36]. Biogenic amine production and hemolytic activity are also harmful to the human body [37]. MJM60958 showed a negative effect with D-lactate, biogenic amine, bile salt deconjugate, and hemolytic activity (Table 2). If LAB are resistant to the antibiotics, the resistant gene may be transferred across the intestinal barrier, but MJM60958 was sensitive to all tested antibiotics (Table 2) and also had strong antibacterial activity against intestinal pathogens which may contribute to intestinal health (Table 4). Moreover, MJM60958 showed high adhesion activity to HT-29 cells and high viability after the OGI test (Table 2 and Table 4). These results suggest that MJM60958 has the characteristics of probiotics and can be used as a probiotic.

We confirmed that MJM60958 significantly reduced body weight and liver weight, and also improved ALT, AST, TG, BUN, and UA index in the blood, which were features of NAFLD observed in the serum and liver of HFD-fed mice (Table 5, Figure 4). Moreover, MJM60958 also reduced serum leptin levels (Figure 4). Similar to insulin, leptin mainly controls food consumption and energy metabolism, further inhibiting hunger and reducing body weight [38]. However, leptin resistance with a high level of leptin in serum may be a feature of steatosis that contributes to the development of NAFLD [39]. In addition, another previous study also showed that serum leptin concentrations are significantly higher in non-alcoholic steatohepatitis patients than in healthy people [40]. Therefore, MJM60958 may improve NAFLD by balancing serum leptin levels.

Liver fat accumulation results from a lack of equilibrium between lipid synthesis and disposal, which are regulated through de novo lipogenesis and fatty acid oxidation [41]. Fat droplets accumulate in the liver when the rate of lipid synthesis is higher than lipid oxidation. H&E staining analysis demonstrated that HFD induced hepatic fat in mice and probiotic treatment reduced fat accumulation (Figure 3). The liver can synthesize new fatty acids from acetyl-CoA by the action of acetyl-CoA carboxylase (ACC) and fatty acid synthase (FAS). Initially, acetyl-CoA carboxylase (ACC) catalysis acetyl-CoA converts to malonyl-CoA and then to palmitate by fatty acid synthase (FAS). Sterol regulatory element-binding protein 1 (SREBP-1), which is activated by insulin and liver X receptor α, is a key transcription factor regulating the transcription of de novo lipogenesis. High blood levels of insulin due to insulin resistance often lead to steatosis in the liver because of SREBP-1 activation, and the suppression of SREBP-1 means protection against the development of a fatty liver. In NAFLD, the expression of ACC and FAS increase in response to the increase of upstream SREBP-1 [42,43,44,45]. ACC is a rate-limiting enzyme that catalyzes acetyl-CoA into malonyl-CoA, which inhibits the expression of carnitine palmitoyltransferase-1A (CPT1A) and consequently blocks mitochondrial fatty acid oxidation [46]. For fatty oxidation, PPARα is a key nuclear receptor that controls the rate of oxidation of fatty acids occurring in the mitochondria and is also related to carnitine palmitoyltransferase-1 (CPT1) [47]. Our results showed that MJM60958 reduced the expression of FAS and ACC, but upregulated the PPARα and CPT1A genes (Figure 5). The Western blotting data in Figure 6 also demonstrates that MJM60958 significantly inhibited the protein expression of FAS, SREBP-1, and strongly increased PPARα protein levels. Our data suggest that MJM60958 improves fatty liver metabolism and reduces NAFLD by downregulating genes and protein expression, which is related to hepatitic lipid synthesis and enhancing the levels of genes and proteins related to hepatitis fatty oxidation.

From 10–20% of NAFLD patients developing NASH, 2–5% of them develop NASH with a hallmark of inflammation. In our study, mice in the HFD group had significantly increased serum IL-1β and slightly induced TNFα levels. With the LAB treatment, the concentration of IL-1β and TNFα in serum strongly declined compared with the HFD treatment (Figure 4). This suggests that MJM60958 consumption can reduce and stop the progression of NAFLD into NASH in mice.

Several studies have demonstrated that the diversity and composition of gut microbiota influenced NAFLD [48,49]. In our research, the diversity of microbiota was decreased in the HFD group and showed increased richness in the MJM60958 treatment. The change was noticeable in the Verrucomicrobia phylum and Akkermansia family, which are known to have functions that contribute to intestinal health and glucose homeostasis and reduce obesity, diabetes, inflammation [50,51,52]. They were absent in the HFD group, but enriched in the LAB treatment (Figure 7A,B). These data suggest that MJM60958 might have the potential to reduce NAFLD through the alteration of the microbiome composition, especially through an increase of Akkermansia and Verrucomicrobia. In addition, SCFAs produced by microorganisms have an important role in NAFLD [53]. Lactic acid is major metabolite produced by Lactobacillus and strains belonging to the Firmicutes phylum. In our research, Firmicutes accounted for 89% to over 97% of the cecal microbiota (Figure 7A), which explains why lactic acid was the main SCFA in feces (Figure 7C). Moreover, lactic acid has been reported to increase in the feces of NAFLD patients [54]. Here, it was increased in the HFD group, and decreased in the MJM60958 group (Figure 7C). Further, acetic acid is also known to upregulate the expression of genes for fatty acid oxidation enzymes in the liver to suppress body fat accumulation [55]. Another study confirmed that the anti-NAFLD effects of astragalus polysaccharides are associated with increased production of acetic acid [56]. In this study, MJM60958 administration also increased acetic acid concentration in mice fed with a HFD which could contribute to the anti-NAFLD effect of MJM60958 (Figure 7D).

## 4. Materials and Methods

### 4.1. Cell Culture

HepG2 cells, a human hepatocellular liver carcinoma cell line, were purchased from the American Type Culture Collection (Rockville, MD, USA). Briefly, cells were cultured in Dulbecco’s Modified Eagle Medium (DMEM) containing 10% fetal bovine serum (FBS), and 1% Penicillin-Streptomycin (10,000 U/mL) purchased from Gibco (Avenue, Waltham, MA, USA) in an incubator at 37 °C under a humidified atmosphere of air containing 5% CO_2_.

HT-29 (human colon adenocarcinoma) cell line was purchased from KCTC (Korea Research Institute of Bioscience and Bioscience Center) and cultured in RPMI 1640 (HyClone Laboratories, Inc., Logan, UT, USA) supplemented with 10% fetal bovine serum (Gibco, Waltham, MA, USA) and 1% streptomycin/penicillin solution (Gibco, Waltham, MA, USA) at 37 °C in 5% CO_2_ atmosphere.

### 4.2. Screening of Probiotic Strains with Inhibitory Activity to Reduce Lipid Accumulation in HepG2 Cells

For screening, 15 LAB strains isolated from fermentation food were tested for their ability to reduce lipid accumulation in an in vitro cell model. The NAFLD cell mode using HepG2 cells stimulated by fatty solution (1 mM Oleic acid, 7.5 µg/mL cholesterol) with 2 layer plates for 6 h was used as in previous studies [57,58] with some modifications. HepG2 cells were seeded at a density of 5 × 10^5^ cells/mL in six-well plates, incubated for 24 h and starved overnight with the FBS-free DMEM medium. After that, 10^8^ to 10^9^ CFU/mL probiotics in a fresh free antibiotic DMEM medium supplemented with fatty solution (1 mM Oleic acid, 7.5 µg/mL cholesterol) were seeded on a Transwell membrane (SPL, Pochon, Korea) and inserted into a six-well culture plate containing HepG2 cells for 6 h. The cells treated with only the fatty solution were considered as a negative control group, and the positive control was based on simvastatin (1 µM). After 6 h of treatment, the transwell membranes were removed and the HepG2 cells were washed and stained.

Oil Red O staining: Cells were washed with phosphate-buffered saline (PBS) and fixed with 10% formaldehyde for 5 min; they were then washed and fixed with 10% formaldehyde for 1 h before staining with 60% Oil Red O diluted in distilled water. After 30 min, the cells were washed three times with distilled water. The dye retained in cells was eluted by adding 100% isopropanol and quantified with a microplate reader at 510 nm.

### 4.3. Assay of Cell Viability

The effect of MJM60958 on the cell viability of HepG2 cells was determined by MTT (3-(4,5-dimethylthiazol-2-yl)-2,5 diphenyl tetrazolium bromide) assay. HepG2 cells were seeded in a 6-well plate at a density of 5 × 10^5^ cells/mL. The cells were incubated, starved, and treated with probiotics for 6 h with the same conditions as before. Following the incubation, the transfer well was removed and the HepG2 cells were treated with 20 µL MTT (5 mg/mL) for 4 h. Finally, the medium was removed and then treated with 200 µL DMSO to dissolve the formazan crystals, the metabolite of MTT. The formazan concentration was measured by a microplate reader at 570 nm (TECAN Spectrofluor Plus, Maennedorf, Switzerland). The cell viability was determined relative to the non-treatment group.

### 4.4. Probiotic Characterization of Candidate LAB Strains

#### 4.4.1. Hemolytic Activity

The hemolytic activity of the *Lactobacillus sakei* MJM60958 was tested following the guideline of the American Society for Microbiology. Briefly, *Lactobacillus sakei* MJM60958 and *Lactobacillus rhamnosus* GG (LGG) as positive control were streaked in tryptic soy agar supplemented with 5% sheep blood (Hardy Diagnostics, A10), and incubated at 37 °C for 24 h. Finally, the hemolytic activity was observed with backlighting. The clear zone surrounding the colony that approached the color and transparency of the base medium was measured to determine hemolytic activity.

#### 4.4.2. D-Lactate Production

The D-lactate production of *Lactobacillus sakei* MJM60958 and LGG (positive control) was tested using a D-lactic acid assay kit (K-DATE, Megazyme, Ireland) according to the protocol supplied by the manufacturer. Briefly, 0.1 mL of LAB culture medium was mixed with 1.5 mL of distilled water, 0.5 mL of supplied buffer (solution 1), 0.1 mL of NAD+ solution (solution 2), and 0.02 mL of D-glutamate-pyruvate transaminase (D-AST) and incubated at room temperature for 3 min. Then, the absorbance of the mixture was measured at 340 nm. Next, 0.02 mL of 2000 U/mL D-lactate dehydrogenases (D-LDH, suspension 4) was added to the above reaction mixture, and the absorbance of the final mixture was measured at 340 nm for 5 min until the D-LDH reaction stopped. D-lactate concentrations were calculated according to the equations according to the manufacturer’s instructions.

#### 4.4.3. Bile Salt Deconjugation

*Lactobacillus sakei* MJM60958 and LGG (positive control) were tested on de Man, Rogosa, and Sharpe (MRS) agar with or without supplemented 0.5% taurodeoxycholic acid sodium salt hydrate (Sigma-Aldrich, Saint Louis, MO, USA) and incubated at 37 °C for 48 h. The halo zone and the opaque white precipitate around the colonies were observed to determine bile salt deconjugation. MRS agar medium plates without supplementation of the conjugated bile acids were used as control.

#### 4.4.4. Biogenic Amine Production

Biogenic amines produced by *Lactobacillus sakei* MJM60958 were determined using a decarboxylase medium supplemented with 1% individual amino acids (L-histidine, L-tyrosine, L-phenylalanine, Arginine, Tryptophan, and L-ornithine) following the previous method [59]. Briefly, 1000 mL decarboxylase medium containing 5 g tryptone, 5 g yeast extract, 5 g beef extract, 2.5 g NaCl, 0.5 g glucose, 1 g tween 80, 0.2 g MgSO_4_, 0.05 g MnSO_4_, 0.04 g FeSO_4_, 2 g ammonium citrate, 0.01 g thiamine, 2 g K_2_HPO_4_, 0.1 g calcium carbonate, 0.05 g pyridoxal 5-phosphate, 0.06 g bromocresol purple, and 20 g agar was adjusted to pH 5.5. The medium without supplemented biogenic amines served as a control. A single colony of selected strains was placed onto agar plates and incubated for 48 h at 37 °C in anaerobic conditions. Purple color formation in medium agar plates with added amino acid, and yellow formation in control agar plates without amino acids were evaluated as amino acid decarboxylase positive.

#### 4.4.5. Mucin Degradation Activity

The mucin degradation activity of MJM60958 was determined using the agarose medium containing 0.3% porcine gastric mucin (Sigma, St. Louis, MO, USA), with or without glucose. Ten microlitres of MJM60958 or *Bacillus* sp. (positive control) were incubated onto the surface of the medium at 37 °C under anaerobic conditions (BBL Gas Pack System, BD) for 3 days. Finally, the culture was stained with 3.5 M acetic acid containing 0.1% amido black for 30 min and destained with 1.2 M acetic acid until the appearance of a clearance zone in the positive control. Mucin degradation activity was determined by measuring the clearance zone around the colony.

#### 4.4.6. Antimicrobial Assay

The antimicrobial activity of MJM60958 and LGG was determined by using the agar well diffusion method. For the antimicrobial test, probiotic strains were inoculated at 37 °C, in MRS broth for 24 h. Then, cell-free culture supernatants of MJM60958 and LGG were harvested by centrifugation, filtered, and added to wells (8 mm diameter) of agar plates containing each pathogenic strain. The pathogenic strains were *Salmonella gallinarum* KCTC 2931, *Escherichia coli* K99, *Escherichia coli* O1 KCTC 2441, *Escherichia coli* 0138, *Escherichia coli* ATCC25922, *Salmonella cholreraesuis* KCTC 2932, *Salmonella typhi* KCTC 2514, and *Pseudomonas aeruginosa* KCCM 11802. The agar plates were incubated at 37 °C for 24 h, and the diameters of inhibition zones except for the well (8 mm) were measured to determine antimicrobial activity. All the tests were performed in triplicate.

#### 4.4.7. Susceptibility of the Candidate Strains to Antibiotics

The two-fold broth microdilution method was used to determine the minimum inhibitory concentration (MIC) with different antibiotics of probiotic strains [60]. The antibiotics used for the test were chloramphenicol, ampicillin, tetracycline, gentamycin, kanamycin, streptomycin, erythromycin, clindamycin, and vancomycin, as determined by the European Food Safety Authority (EFSA, 2012) [61]. The MIC cutoff values for various antibiotics recommended by the EFSA for facultative heterofermentative Lactobacillus were used to determine the antibiotic susceptibility of *Lactobacillus sakei* MJM60958 after 48 h treatment.

#### 4.4.8. Oro-Gastrointestinal Transit Assay

To check the viability of selected trains challenged by several sources of stress such as low pH, bile, and digestive enzyme, an oro-gastro-intestinal (OGI) transit assay was used [62]. Briefly, 10^9^ CFU/mL of *Lactobacillus sakei* MJM60958 and LGG (positive control) were initially treated with oral stress solution (NaCl, 0.62 g/100 mL; KCl, 0.22 g/100 mL; CaCl_2_, 0.022 g/100 mL; NaHCO_3_, 0.12 g/100 mL; lysozyme 0.015 g/100 mL) at 37 °C for 10 min. After centrifugation at 1800× *g* for 5 min, the cells were resuspended in gastric stress solution (0.3% pepsin, adjusted to pH 3) for 30 min, then subjected to gastric stress solution at pH 2 for an additional 30 min. Finally, the gastric solution was replaced by intestinal solution (NaCl, 0.5 g/100 mL; KCl, 0.06 g/100 mL; CaCl_2_, 0.025 g/100 mL; pancreatin, 0.1 g/100 mL; bile bovine, 0.3 g/100 mL; pH 7), and the cells were treated for 2 h. During this process, selected strains subjected to PBS instead of the stress solution were used as a control. Cell viability at each step was determined by plating the diluted cells in MRS medium, and counting and calculating the number of colonies after 48 h.

#### 4.4.9. Adhesion to HT-29 Cells

HT-29 cells were seeded at 2 × 10^5^ cells/well in a 24-well plate and cultured at 37 °C for 24 h. HT-29 monolayers were washed three times with PBS, and the medium was replaced with antibiotic-free RPMI 1640. Bacterial strains were inoculated into each well at approximately 10^8^ CFU/mL and cultured at 37 °C for 2 h. Non-adhered bacteria were removed by washing thrice with PBS, followed by the addition of 1 mL of Trypsin (Sigma-Aldrich) into each well and incubated at 37 °C for 5 min. Incubated cells were harvested from each well, and adherent bacterial cells were diluted and plated on MRS agar and incubated at 37 °C for 48 h. Adhesion activity was calculated by dividing the remaining bacteria grown on MRS agar by the initial inoculation bacteria. All the experiments were performed in triplicate.

### 4.5. Animal Study

This experiment was performed using C57BL/6 male mice (7-weeks old) purchased from Raon Bio (Daejeon, Korea). The mice were separated randomly into groups of 4 mice in each cage and maintained under a 12 h light-dark cycle at a constant temperature and humidity (22 ± 2 °C and 55 ± 5%), without access to food and water for 1 week to acclimatize. The procedure for the animal experiments was approved by the Ethics committee of Animal Experiments of the Myongji University (MJIACUC-2021003) and performed following the NIH guidelines for laboratory animals.

#### 4.5.1. Experimental Design

Previous studies have shown that feeding with a high-fat diet (HFD) composed of 45% fat, 35% carbohydrates, and 20% protein for 12 weeks can induce NAFLD, as shown by increased lipid accumulation, and develop steatosis in C57BL/6 mice [63,64]. Based on these studies, we used C57BL/6 mice fed with HFD to create animal models of non-alcoholic fatty liver disease.

After acclimatization for 1 week, the mice were divided into five groups for the 12-week treatments (Figure 8): (1) control group (control), fed with a normal diet (Raon Bio, Daejeon, Korea); (2) high-fat diet group (HFD) as a negative control, fed with a high-fat diet (Raon Bio, Yongin, Korea) containing 45% fat, 35% carbohydrate, and 20% protein; (3) silymarin (positive control; Sigma-Aldrich, Saint Louis, MO, USA), fed with HFD and silymarin provided orally (50 mg/kg/day); (4) MJM60958 low dose (MJM60958 (10^8^)), fed with HFD and receiving 10^8^ CFUs/day of *L. sakei* MJM60958 for 12 weeks by oral gavage; (5) MJM60958 high dose (MJM60958 (10^9^)), fed with HFD and receiving 10^9^ CFUs/day of *L. sakei* MJM60958 for 12 weeks by oral gavage.

Every day, *L. sakei* MJM60958 was freshly cultured, harvested, washed 2 times with PBS, and resuspended in distilled water for treatment. During the experimental process, the food intake and body weight were measured twice per week. After 12 weeks of treatment, the mice were bled and euthanized after anesthesia induction with 3% isoflurane by cervical dislocation, and the organs were weighed, sectioned, and then fixed with 10% formalin, sucrose, or liquid nitrogen for further analysis.

#### 4.5.2. Serum Biochemical Analysis

Blood samples were immediately kept on ice for 1 h after bleeding and then centrifuged at 2000× *g* for 10 min at 4 °C to collect serum. The serum was separated and stored at −80 °C until analyses. The levels of serum alanine aminotransferase (ALT), triglyceride (TG), serum aspartate aminotransferase (AST), total cholesterol (T-CHO), creatinine (CRE), blood urea nitrogen (BUN), and serum uric acid (UA) were measured using a biochemical blood analyzer (FUJIFILM DRI-CHEM NX500i, Tokyo, Japan).

#### 4.5.3. Immunoassay for the Plasma Levels of Leptin, Adiponectin, IL1β, and TNFα

The serum levels of leptin, adiponectin, IL1β, and TNFα in mice were measured using the mouse ELISA enzyme immunoassay kit (Abcam, Waltham, MA, USA) following the protocol provided by the manufacturer.

#### 4.5.4. Histopathological Observation

Histological examination of mice livers was performed using the hematoxylin-eosin (H&E) staining method. The liver tissues, which were fixed with 10% formalin, were embedded in paraffin, sliced into 5-μm-thick sections, stained with H&E by the standard protocol, and the images were analyzed by a computer image analysis system at 400× magnification (CaseViewer by 3DHITECH, Budapest, Hungary). The images were visualized and graded on 4 levels based on the percentage of fat vacuolation in hepatocytes: level 0 (healthy, <5%), level 1 (mild, 5–33%), level 2 (moderate, 34–66%), and level 3 (severe, >66%) [65].

#### 4.5.5. Fecal Sample Analysis

After sacrifice, the intestine was immediately dissected, and the feces were collected in a tube and stored at −80 °C before use. For metagenome analysis, DNA was extracted with the ExgeneTM Stool DNA mini kit (GeneAll, Seoul, Korea) according to the manufacturer’s protocol. The V3–V4 region of the bacterial 16S rRNA gene was amplified using barcoded universal primers 341F and 805R.

Microbiome profiling was conducted with the 16S-based Microbial Taxonomic Profiling platform of EzBioCloud Apps (Sanigen, Anyang, Korea).

#### 4.5.6. RNA Extraction and Reverse Transcription-Quantitative Polymerase Chain Reaction (RT-qPCR) Analysis

Trizol reagents were used to extract RNAs from liver tissue. Firstly, mouse liver tissue was ground in liquid nitrogen, transferred to an e-tube and stored in an icebox. Then, 1 mL trizol was added to each sample and mixed. Next, 200 μL of chloroform was added to each e-tube and the samples were centrifuged at 14,000 rpm for 15 min at 4 °C. After centrifugation, the upper, aqueous phase was transferred into a fresh e-tube and mixed with 500 μL isopropanol. The samples were then centrifuged at 14,000 rpm for 10 min, and the RNA pellet was collected and washed with 1 mL ethanol 75%. Finally, the RNA pellet was dissolved in RNAse-free water and the RNA concentration was determined at OD 260 nm.

Reverse transcription was conducted with a TaKaRa Kit and 1 µg of total RNA was used. The cDNA samples were then synthesized using a Primescript 1st Strand cDNA Synthesis Kit (Takara Bio, Inc., Otsu, Japan) following the manufacturer’s protocol. The reaction conditions were 42 °C for 60 min and 95 °C for 5 min. qPCR was performed with an SYBR Green Master Mix (Bio-Rad Laboratories, Inc., Otsu, Japan). Quantitative real-time PCR was performed for fatty acid synthase (FAS), acetyl-CoA carboxylase (ACC), peroxisome proliferator-activated receptor alpha (PPARα), carnitine palmitoyltransferase 1a (CPT1A), and interleukin 6 (IL-6). The expression levels of the target genes were normalized to β-actin. In all assays, FAS, ACC, PPARα, CPT1A, and IL-6 cDNA were amplified using a program as follows: stage 1, 94 °C for 10 min; stage 2, 45 repetitions of 94 °C for 15 s, 60 °C for 1 min; and stage 3, 95 °C for 10 s, 65 °C for 60 s, 97 °C for 1 s. Primers for RT-PCR were: forward (AGGGGTCGACCTGGTCCTCA), reverse (GCCATGCCCAGAGGGTGGTT) for FAS; forward (AACATCCCGCACCTTCTTCTAC), reverse (CTTCCACAAACCAGCGTCTC) for ACC; forward (AGAGCCCCATCTGTCCTCTC), reverse (ACTGGTAGTCTGCAAAACCAAA) for PPARα; forward (TGGCATCATCACTGGTGTGTT), reverse (GTCTAGGGTCCGATTGATCTTTG) for CPT1A; forward (ACAACCACGGCCTTCCCTACTT), reverse (CACGATTTCCCAGAGAACATGTG) for IL-6; and forward (ACAACCACGGCCTTCCCTACTT), reverse (CACGATTTCCCAGAGAACATGTG) for β-actin.

#### 4.5.7. Western Blot Analysis

Protein expression levels of liver FAS, sterol regulatory element-binding protein 1 (SREBP-1), PPARα, and β-actin as an internal control were determined using Western blotting. Total liver tissue protein was extracted from frozen liver samples. Firstly, frozen liver samples were ground with liquid nitrogen to powder. Then, RIPA lysis buffer supplemented with 1% protease and phosphatase inhibitors was added to each sample and incubated for 30 min on ice. After that, the lysates were centrifuged at 14,000 rpm for 10 min at 4 °C, the supernatants were collected and the protein concentration was measured using a BCA protein assay kit (Thermo Scientific, Rockford, IL, USA). Then, 40 µg of protein of each sample was loaded on a 10% sodium dodecyl sulfate-polyacrylamide gel electrophoresis (SDS-PAGE) gel and transferred to polyvinylidene difluoride (PVDF) membranes. The membranes were blocked with Tris-buffered saline (TBS) containing 3% BSA (Bovine Serum Albumin) for 1 h at 4 °C. Then, the membranes were incubated with FAS antibody (1:200, Abcam, Waltham, MA, USA), SREBP-1 antibody (1:1000, Abcam, Waltham, MA, USA), PPARα antibody (1:1000, Invitrogen, Waltham, MA, USA), or β-actin antibody (1:1000, Abcam, Waltham, MA, USA) overnight. Following washing 3 times with Tris-buffered saline containing 0.1% (*v*/*v*) tween-20 (TBS-T) and TBS, the membranes were incubated with horseradish peroxidase (HRP)-conjugated secondary antibody (1:10,000, Abcam, Waltham, MA, USA) at 4 °C for 1 h. After washing, immunoreactive bands were enhanced using chemiluminescence reagents, then the membrane photos were captured. Finally, the ImageJ program was used for quantifying protein expression. All the experiments were performed in triplicate.

#### 4.5.8. Statistical Analysis

Data from all the experiments were analyzed using Prism software (Prism version 7; GraphPad Software, San Diego, CA, USA). One-way ANOVA or student’s *t*-test was used to compare the means of the experimental groups. *p* values < 0.05 were considered statistically significant.

## 5. Conclusions

In this study, *Lactobacillus sakei* MJM60958 showed the characteristic of probiotics, reduced lipid accumulation in hepatic cells and alleviated non-alcoholic fatty liver disease in a NAFLD mice model. Administration of *Lactobacillus sakei* MJM60958 reduced body weight and liver weight and attenuated the NAFLD-related biomarkers, such as ALT, AST, TG, BUN, and UA, and reduced hepatic damage in the NAFLD mice model. Moreover, *Lactobacillus sakei* MJM60958 suppressed lipid accumulation in liver mice by inhibiting the expression of genes and proteins related to lipid accumulation and upregulating genes and protein levels of lipid oxidation. Further, the composition of the gut microbiota was changed and the proportion of the *Akkermansiaceae* family was increased. The concentration of SCFAs such as acetic acid, which is known to reduce NAFLD, was increased. Therefore, *Lactobacillus sakei* MJM60958 may be used as a potential probiotic to treat and prevent NAFLD.

## Figures and Tables

**Figure 1 ijms-23-13436-f001:**
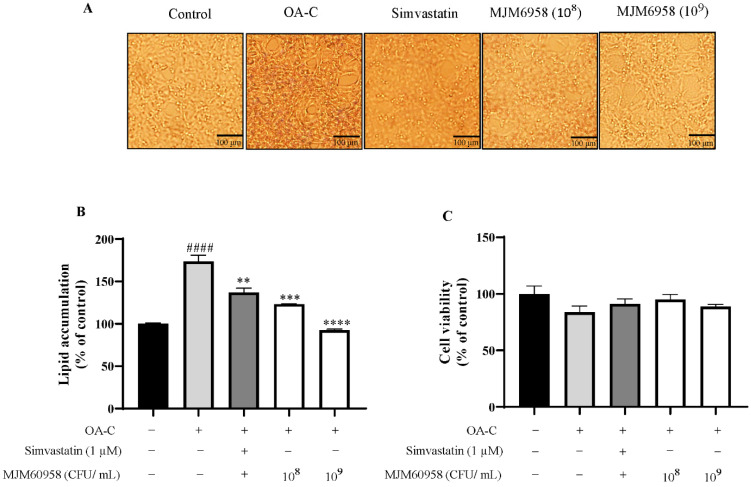
Effect of MJM60958 on HepG2 cells. (**A**) Typical image of Oil red O staining, scale bar: 100 μm. (**B**) The effect of MJM60958 on the lipid accumulation of HepG2 cells. HepG2 cells were stimulated with 1 nM oleic acid and 7.5 µg/mL cholesterol (OA-C) and treated with 10^8^ and 10^9^ CFUs/mL of MJM60958 for 6 h. Simvastatin (1 µM) was used as a positive control. At the end of treatment, the cells were stained with Oil red O reagent and lipid accumulation was quantified by dissolving the stained fat droplet in isopropanol and measurement at 510 nm by an ELISA reader (TECAN Spectrofluor Plus, Maennedorf, Switzerland). (**C**) The effect of MJM60958 on the viability of HepG2 cells was measured using an MTT assay. Results are presented as the mean ± standard deviation of triplicate independent experiments. ^####^ *p* < 0.01 compared with the control; ** *p* < 0.01, *** *p* < 0.001, **** *p* < 0.0001 compared with the OA-C treatment.

**Figure 2 ijms-23-13436-f002:**
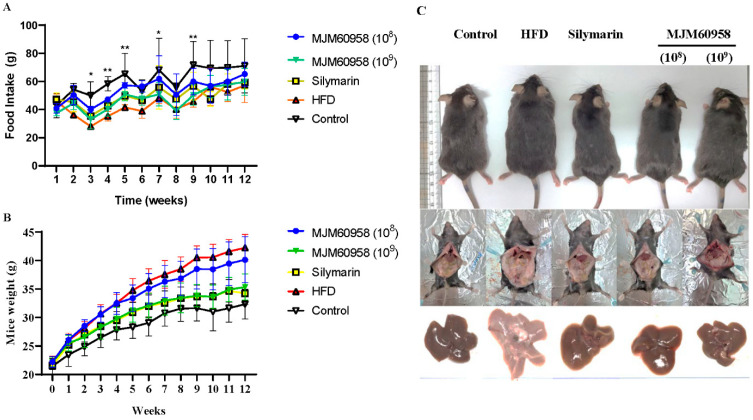
Food intake, body weight during 12 weeks, and mice images at the end of week 12. (**A**) Food intake and (**B**) body weights were measured every week. (**C**) Mouse and liver images after 12 weeks of treatment. Data were analyzed by two-way ANOVA (*n* = 8 mice per group). * *p* < 0.05, ** *p* < 0.01 compared with HFD.

**Figure 3 ijms-23-13436-f003:**
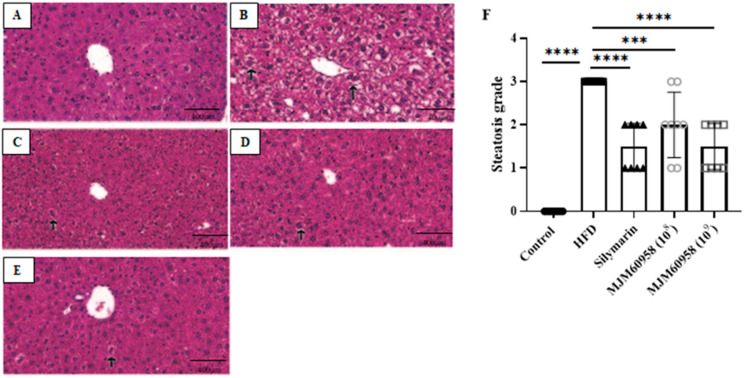
Effect of MJM60958 on histopathology of the liver sections stained with H&E. Black arrow indicates ballooned liver cells. Photographs of HE-stained sections of livers: (**A**) Control, (**B**) HFD (**C**) Silymarin, (**D**) MJM60958 (10^8^), and (**E**) MJM60958 (10^9^). (**F**) steatosis grade was quantified based on the percentage of fat vacuolation in hepatocytes: level 0 (healthy, <5%), level 1 (mild, 5–33%), level 2 (moderate, 34–66%), and level 3 (severe, >66%). Data are presented as the mean ± SD (*n* = 8), *** *p* < 0.001, **** *p* < 0.0001 compared with HFD group. Scale bar: 400 μm. (Magnification, 400×).

**Figure 4 ijms-23-13436-f004:**
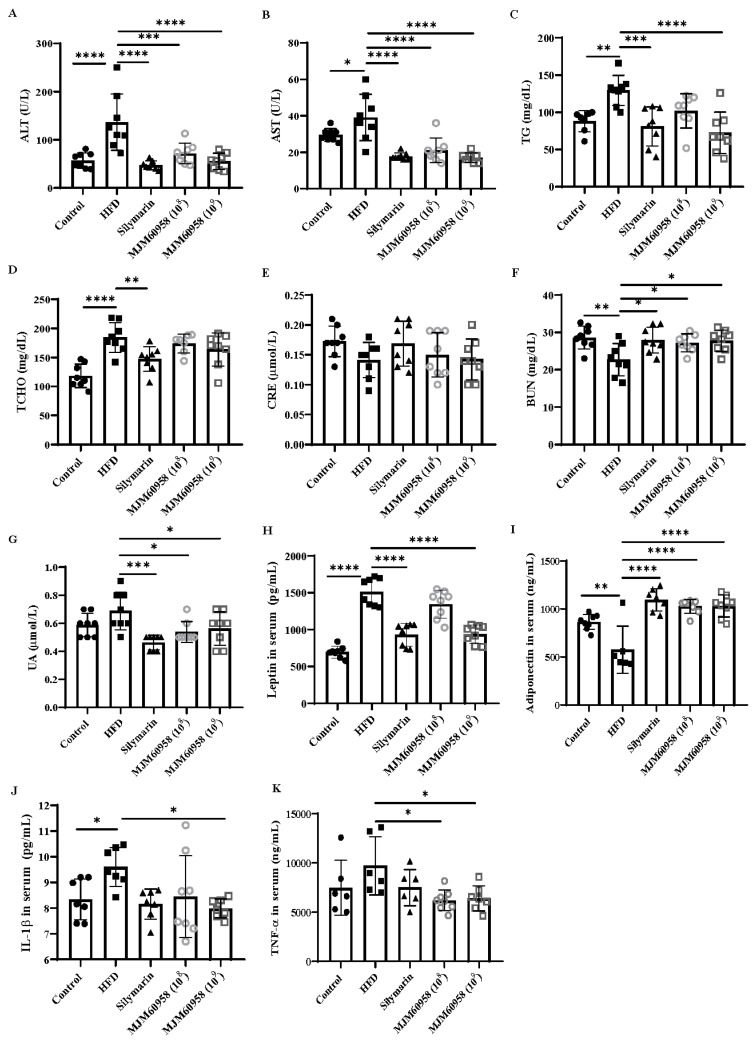
Effects of MJM60958 or Silymarin on the serum levels of (**A**) ALT, (**B**) AST, (**C**) TG, (**D**) TCHO, (**E**) CRE, (**F**) BUN, (**G**) UA, (**H**) leptin, (**I**) adiponectin, (**J**) IL-1β and (**K**) TNF-α in mice fed HFD for 12 weeks. Values are the means ± SD (*n* = 6 to 8 per group). * *p* < 0.05, ** *p* < 0.01, *** *p* < 0.001, **** *p* < 0.0001, compared with HFD group.

**Figure 5 ijms-23-13436-f005:**
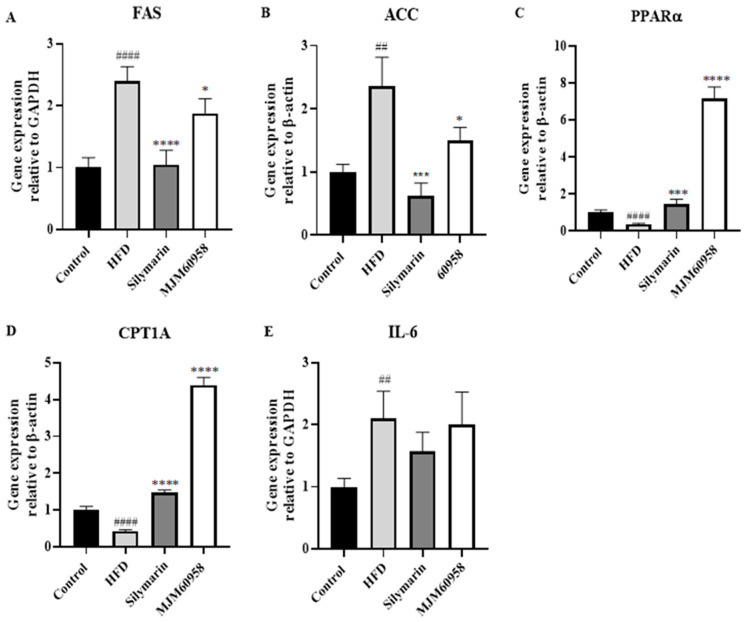
Effect of *L. sakei* MJM60958 on the mRNA expression of liver tissues in mice after 12 weeks of treatment. mRNA of (**A**) FAS, (**B**) ACC, (**C**) PPARα, (**D**) CPT1A, and (**E**) IL-6 were quantified using qRT-PCR. Data are expressed as the means ± standard deviation of triplicate independent experiments. ^##^ *p* < 0.01, ^####^ *p* < 0.0001 compared with control group; * *p* < 0.05, *** *p* < 0.001, **** *p* < 0.0001 compared with HFD group.

**Figure 6 ijms-23-13436-f006:**
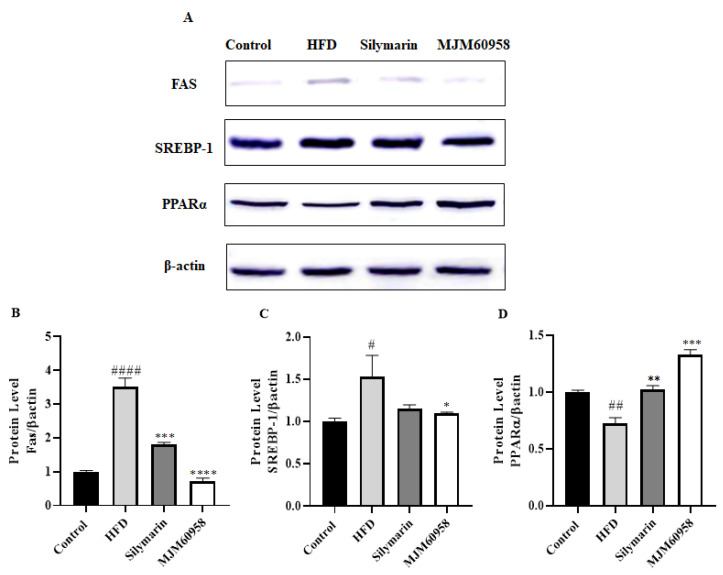
Effect of MJM60958 on FAS, SREBP-1, and PPARα protein levels in liver tissues of mice fed either HFD, *L. sakei* MJM60958, or silymarin for 12 weeks. (**A**) Protein expression was determined by Western blotting and protein expression of (**B**) fatty acid synthase/β-actin, (**C**) Sterol regulatory element-binding protein 1/β-actin, (**D**) peroxisome proliferator-activated receptor alpha/β-actin were quantified using the ImageJ program. Data are expressed as the means ± standard deviation of triplicate independent experiments. ^#^ *p* < 0.05, ^##^ *p* < 0.01, ^####^ *p* < 0.0001 compared with control group; * *p* < 0.05, ** *p* < 0.01, *** *p* < 0.001, **** *p* < 0.0001 compared with HFD group.

**Figure 7 ijms-23-13436-f007:**
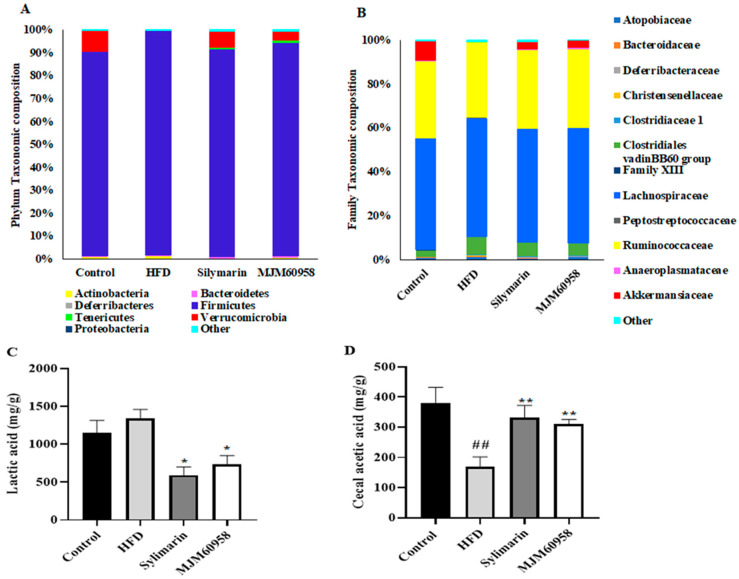
Effect of MJM60958 on the composition of cecal microbiota and SCFAs content in cecal. (**A**) The structure of the gut microbiota at the phylum level. (**B**) The structure of the gut microbiota for HFD at the genus level. Short-chain fatty acids (SCFAs) content in cecal was measured by gas chromatograph: (**C**) acetic acid, and (**D**) lactic acid. Data are expressed as the means ± SD (*n* = 3). ^##^ *p* < 0.01 compared with control group; * *p* < 0.05, ** *p* < 0.01 compared with HFD group.

**Figure 8 ijms-23-13436-f008:**
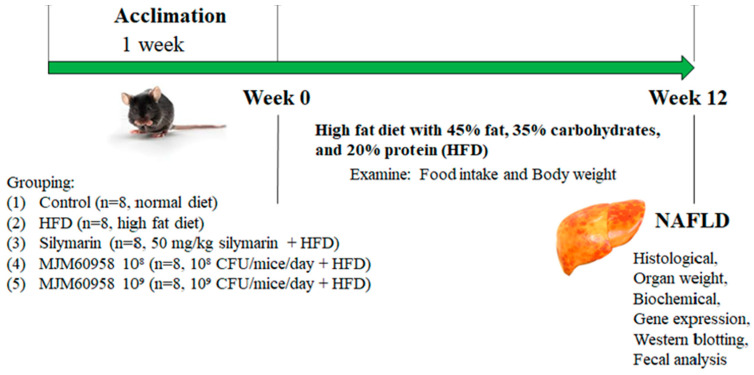
Animal experiment design.

**Table 1 ijms-23-13436-t001:** Assessment of ability to inhibit lipid accumulation on HepG2 cells of candidate strains.

Samples	Lipid Accumulation (% of Control)	Percent Reduction (Compared with OA-C)
Control	100 ± 1.12	-
OA-C	173.76 ± 7.18 ^####^	-
Simvastatin (1 µM)	136.87 ± 5.23 **	21.23
*Lactobacillus gasseri* MJM61024	122.59 ± 0.85 ***	29.45
*Lactobacillus rhamnosus* GG	151.53 ± 2.12 **	12.79
*Lactobacillus rhamnosus* MJM60370	157.33 ± 5.47 *	8.93
*Lactobacillus brevis* MJM60386	132.25 ± 1.23 **	23.89
*Lactobacillus heveticus* MJM60419	158.27 ± 4.87 *	8.91
*Lactobacillus fermentum* MJM60430	172.69 ± 4.23	0.62
*Lactobacillus casei* MJM60432	177.36 ± 5.06	−2.07
*Lactobacillus paracasei* MJM60434	152.72 ± 5.97 *	12.11
*Lactobacilus helveticus* MJM60463	160.42 ± 4.37	7.68
*Streptococus thermophilus* MJM60636	171.11 ± 2.98	1.53
*Lactobacillus sakei* MJM60958	92.46 ± 1.16 ****	46.79
*Lactobacillus rhamnosus* MJM60711	146.26 ± 3.19 **	15.83
*Lactobacilus helveticus* MJM60825	152.72 ± 5.97 *	12.11
*Lactobacillus plantarum* MJM61025	145.53 ± 3.08 **	16.25
*Lactobacillus curvatus* MJM61026	139.46 ± 3.39 **	19.74

OA-C: 1 nM oleic acid and 7.5 µg/mL cholesterol (negative control). ^####^ *p* < 0.0001 compared with control; * *p* < 0.05, ** *p* < 0.01, *** *p* < 0.001, **** *p* < 0.0001 compared with OA-C.

**Table 2 ijms-23-13436-t002:** Safety assessment of *L. sakei* MJM60958.

Safety Test	*L. sakei* MJM60958	LGG
	(MJM60958)	
Antibiotics *		
Ampicillin	1	1
Vancomycin	256 (NR)	512 (NR)
Gentamycin	16	32 (R)
Kanamycin	2	R
Streptomycin	1	32 (R)
Tetracycline	1	1
Clindamycin	1	1
Erythromycin	1	1
Chloramphenicol	1	4
D-lactate production	−	−
Bile salt deconjugation	−	−
Bioamine production	−	−
L-Histidine	−	−
L-Tyrosine	−	−
L-phenylalanine	−	−
Arginine	−	−
Tryptophan	−	−
L-ornithine	−	−
Mucin degradation	−	−
Hemolytic activity	−	−
Adhesion to HT-29 (%)	5.09 ± 0.29	3.19 ± 0.1

NR: not required, R: resistant, − no activity, * MIC value for the antibiotics recommended by European food safety authority (EFSA), 2012.

**Table 3 ijms-23-13436-t003:** Oro-gastrointestinal transit assay of *L. sakei* MJM60958 and *L. rhamnosus* GG.

OGI Transit		(Log_10_CFU/mL)	
		MJM60958	LGG
Initial		9.05 ± 0.045	9.12 ± 0.165
Oral stress	−	9.04 ± 0.05	9.09 ± 0.039
	+	9.04 ± 0.075	9.00 ± 0.006
Gastric stress (pH3)	−	9.03 ± 0.054	9.07 ± 0.026
	+	8.86 ± 0.03 *	8.84 ± 0.052 *
Gastric stress (pH2)	−	9.04 ± 0.045	9.05 ± 0.007
	+	8.44 ± 0.06 **	8.74 ± 0.061 *
Intestinal stress	−	9.03 ± 0.109	9.08 ± 0.092
	+	7.16 ± 0.004 ***	7.79 ± 0.095 *

− without stress, + with stress. * *p* < 0.05, ** *p* < 0.01, *** *p* < 0.001 compared with no stress.

**Table 4 ijms-23-13436-t004:** Antibacterial activity of *L. sakei* MJM60958 and LGG against intestinal pathogens.

Strains	Diameter of Zone Inhibition (mm)	
	*L. sakei* MJM60958	LGG
*Salmonella gallinarum* KCTC 2931	10	10
*Escherichia coli* K99	4	8
*Escherichia coli* O1 KCTC 2441	8	8
*Escherichia coli* 0138	6	6
*Escherichia coli* ATCC25922	6	8
*Salmonella chloeraesuis* KCTC 2932	10	8
*Salmonella typhi KCTC* 2514	6	8
*Pseudomonas aeruginosa* KCCM 11802	10	10

**Table 5 ijms-23-13436-t005:** Effect of *L. sakei* MJM60958 on body weight and organs (*n* = 8) after 12 weeks of treatment.

Group	Control	HFD	Silymarin	MJM60958 (10^8^)	MJM60958 (10^9^)
Bodyweight (g)	30.35 ± 1.86 ****	42.93 ± 2.36	33.78 ± 2.45 ****	39.89 ± 3.91	34.14 ± 1.83 ****
Epidydimal fat weight (g)	0.22 ± 0.045 ***	0.34 ± 0.063	0.3 ± 0.047	0.33 ± 0.084	0.31 ± 0.061
Kidney weight (g)	0.18 ± 0.017	0.2 ± 0.019	0.18 ± 0.021	0.2 ± 0.024	0.17 ± 0.028
Liver weight (g)	1.28 ± 0.17 ****	1.77 ± 0.30	1.20 ± 0.13 ****	1.30 ± 0.22 ****	1.24 ± 0.11 ****
Liver weight/Body weight (%)	4.09 ± 0.40	4.44 ± 0.25	3.73 ± 0.26 *	3.29 ± 0.71 ****	3.63 ± 0.38 **

* *p* < 0.05, ** *p* < 0.01, *** *p* < 0.001, **** *p* < 0.0001 compared with HFD.

## Data Availability

The raw data used to support the findings of this study will be made available by the authors, without undue reservation, to any qualified researcher.

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
