# Peer review of "Lactobacillus sakei MJM60958 as a Potential Probiotic Alleviated Non-Alcoholic Fatty Liver Disease in Mice Fed a High-Fat Diet by Modulating Lipid Metabolism, Inflammation, and Gut Microbiota"

_ijms, 2022, doi:10.3390/ijms232113436_

Round 1
Reviewer 1 Report
This work studied the effect of a Lactobacillus strain, MJM60958 which was isolated from fermented food, on the attenuation of non-alcoholic fatty liver disease in mice. The study was well designed and the data was solid. The manuscript was well written, and I have some minor comments as below:
Section 2.1, MJM60958 treatment at both concentrations 10⁸ and 10⁹ CFU/mL no toxicity and slightly à showed no toxicity ……
Figure 1A, the picture is not clear enough to figure out the cell morphology and red color of dye, please provide high resolution picture.
Figure 1A. figure legend, please indicate p valur for ####, and ****. And “p” should be italic according to the journal style.
Table 3, please give statistical analysis between stress and no stress conditions.
Author Response
Dear reviewers,
Thank you for your careful review on our manuscript. Your comments helped us improve the quality of the draft. We revised the contents according to your comments. We hope our revision are clear enough to answer your questions. The point-to-point response to the reviewer’s comments are given below.
Reviewer 1:
This work studied the effect of a Lactobacillus strain, MJM60958 which was isolated from fermented food, on the attenuation of non-alcoholic fatty liver disease in mice. The study was well designed and the data was solid. The manuscript was well written, and I have some minor comments as below:
Section 2.1, MJM60958 treatment at both concentrations 10⁸ and 10⁹ CFU/mL no toxicity and slightly à showed no toxicity ……
- This sentence was corrected.
Figure 1A, the picture is not clear enough to figure out the cell morphology and red color of dye, please provide high resolution picture
- A higher resolution picture was provided, and this figure was rearranged.
Figure 1A. figure legend, please indicate p valur for ####, and ****. And “p” should be italic according to the journal style
- This was corrected.
Table 3, please give statistical analysis between stress and no stress conditions.
- We added statistical analysis between stress and no stress conditions.
Reviewer 2 Report
I enjoyed reviewing this interesting experimental study. Using a very elegant animal model, the authors observed that Lactobacillus sakei MJM60958 acts as a probiotic, reducing lipid accumulation in liver cells and alleviating non-alcoholic fatty liver disease in NAFLD mice.
The study is well written, methodologically correct and elegant. The tables and figures are clear.
I only suggest that authors include some more up-to-date references in the text and in the bibliography.
In particular, NAFLD and insulin-resistance are bidirectionally correlated. Two very recent reviews explain in an updated and complete way the pathophysiological mechanisms that support this relationship (Antioxidants, 2021, 10 (2), pp. 1–25, 270. doi: 10.3390 / antiox10020270. - Processes, 2021, 9 (1 ), pp. 1–18, 135. doi: 10.3390 / pr9010135). These references should be added in discussion and bibliografy.
Author Response
Dear reviewers,
Thank you for your careful review on our manuscript. Your comments helped us improve the quality of the draft. We revised the contents according to your comments. We hope our revision are clear enough to answer your questions. The point-to-point response to the reviewer’s comments are given below.
I enjoyed reviewing this interesting experimental study. Using a very elegant animal model, the authors observed that Lactobacillus sakei MJM60958 acts as a probiotic, reducing lipid accumulation in liver cells and alleviating non-alcoholic fatty liver disease in NAFLD mice.
The study is well written, methodologically correct and elegant. The tables and figures are clear.
I only suggest that authors include some more up-to-date references in the text and in the bibliography.
In particular, NAFLD and insulin-resistance are bidirectionally correlated. Two very recent reviews explain in an updated and complete way the pathophysiological mechanisms that support this relationship (Antioxidants, 2021, 10 (2), pp. 1–25, 270. doi: 10.3390 / antiox10020270. - Processes, 2021, 9 (1 ), pp. 1–18, 135. doi: 10.3390 / pr9010135). These references should be added in discussion and bibliografy.
- Thank you for your careful review on our manuscript, and we deeply appreciate your suggestion. Since we didn't show the insulin level in our data, we will refer these reference in our further study.